# CoQ_10_ and Resveratrol Effects to Ameliorate Aged-Related Mitochondrial Dysfunctions

**DOI:** 10.3390/nu14204326

**Published:** 2022-10-16

**Authors:** Gaia Gherardi, Giovanni Corbioli, Filippo Ruzza, Rosario Rizzuto

**Affiliations:** 1Department of Biomedical Sciences, University of Padova, 35131 Padova, Italy; 2Solgar Italia Multinutrient Spa, Via Prima Strada 23/3, 35129 Padova, Italy

**Keywords:** mitochondria, resveratrol, coenzymeQ10, aging

## Abstract

Mitochondria participate in the maintenance of cellular homeostasis. Firstly, mitochondria regulate energy metabolism through oxidative phosphorylation. In addition, they are involved in cell fate decisions by activating the apoptotic intrinsic pathway. Finally, they work as intracellular signaling hubs as a result of their tight regulation of ion and metabolite concentrations and other critical signaling molecules such as ROS. Aging is a multifactorial process triggered by impairments in different cellular components. Among the various molecular pathways involved, mitochondria are key regulators of longevity. Indeed, mitochondrial deterioration is a critical signature of the aging process. In this scenario, we will focus specifically on the age-related decrease in CoQ levels, an essential component of the electron transport chain (ETC) and an antioxidant, and how CoQ supplementation could benefit the aging process. Generally, any treatment that improves and sustains mitochondrial functionality is a good candidate to counteract age-related mitochondrial dysfunctions. In recent years, heightened attention has been given to natural compounds that modulate mitochondrial function. One of the most famous is resveratrol due to its ability to increase mitochondrial biogenesis and work as an antioxidant agent. This review will discuss recent clinical trials and meta-analyses based on resveratrol and CoQ supplementation, focusing on how these compounds could improve mitochondrial functionality during aging.

## 1. Introduction

Mitochondria play a crucial role in controlling energy metabolism. Indeed, ATP is produced within the organelle. Oxygen is the limiting step of this process. However, in the absence of oxygen, cytoplasmatic anaerobic respiration allows the oxidation of glycolytic products less efficiently.

The mechanism that deals with ATP production is the electron transport chain (ETC) formed by respiratory chain complexes I-IV, which can transfer electrons stepwise to reduce oxygen eventually. Glycolysis, fatty acid oxidation, and tricarboxylic acid cycle (TCA) produce NADH and FADH_2_, which are energy-rich molecules able to donate electrons to the ETC. Notably, during this process, protons are pumped out of the mitochondrial matrix in the intermembrane space, producing negative membrane potential inside (−180 mV) [1,2]. The proton motive force, defined as the chemiosmotic potential generated by the proton gradient, drives ADP phosphorylation through the ATP synthase (complex V). Noteworthy, ETC and ATP synthase are tightly coupled. Thus, ATP synthase inhibition results in the block of ETC and cellular respiration [3].

The ETC is the primary source of reactive oxygen species (ROS) production; indeed, although oxidative phosphorylation (oxphos) is an efficient process, a small number of electrons leak from the ETC and interact with molecular oxygen to produce ROS [4]. This scenario suggests that mitochondria should be the first target of oxidative damage supporting the so-called “free radical theory of aging” [5]. Cells possess different antioxidant mechanisms based on enzyme activities and antioxidant agents. Antioxidant enzymes consist of superoxide dismutase and other peroxidases such as catalase, glutathione peroxidase, thioredoxin reductase, and peroxiredoxin. In addition, antioxidant molecules include carotenoids, vitamins C and E, α-lipoic acid, glutathione, flavonoids, and the reduced form of Coenzyme Q_10_ (CoQH_2_) [6].

Mitochondria are not only master regulators of energy metabolism but also key players in a variety of cellular signaling pathways involved in many different physiopathological conditions, e.g., stress response, cell death, and ROS signaling.

Given the fundamental role of mitochondria, it is not surprising that they are involved in the aging process. Notably, aging is accompanied by a decline in mitochondrial mass and function in different tissues. Healthy mitochondria maintenance is essential to counteract this process; thus, metabolites and compounds able to sustain mitochondrial homeostasis are good candidates to ameliorate age-related mitochondrial decline [7,8]. Furthermore, emerging evidence suggests that natural active compounds could be used in different therapeutic applications [9]. In particular, nutraceuticals could have beneficial effects to counteract the age-related decline.

Here, we review the role of mitochondria during aging and how resveratrol, a natural compound, and CoQ_10_, an essential lipid with a dual role as a key electron transfer factor and an antioxidant, could exert a beneficial effect on maintaining mitochondrial homeostasis, specifically during aging.

## 2. Mitochondrial Alterations during Aging

Several studies show that aging is accompanied by a decrease in mitochondrial function, which is supposed to contribute to general tissue decline [10]. This scenario has been demonstrated in model organisms as well as in humans. Specifically, the age-related degeneration in mitochondrial functionality is well documented; thus, for a comprehensive review of this topic, see [7]. This section will summarize different aspects of mitochondrial biology during aging, such as mtDNA mutations, ROS production, mitochondrial biogenesis, and mitohormesis.

Mitochondria become aberrant during aging. In terms of morphology and quantity, they become fewer and enlarged with cristae abnormalities [11]. Obviously, these morphological alterations affect mitochondrial functionality. Indeed, a decrease in mitochondrial respiratory chain enzyme activities and membrane potential is shown during aging, leading to reduced ATP production [12] (Figure 1). In addition, dysfunctional proteins and increased mtDNA mutations result from the high oxidative damage.

### 2.1. Re-Evaluation of the “Mitochondrial Free Radical Theory of Aging”

As aforementioned, the mitochondrial free radical theory of aging suggests that mitochondrial alterations during aging trigger increased ROS production, which, in turn, boosts mitochondrial dysfunctions [13]. Consistent literature supports this theory; however, this concept has been re-evaluated recently, starting from unexpected results [14]. Firstly, it has been demonstrated that enhanced ROS production extends the life span of worms and yeast [15,16,17]. Furthermore, the increased oxidative metabolism in mice is insufficient to trigger premature aging, although it enhances cancer incidence [18,19]. In addition, genetic manipulations aimed at increasing antioxidant defenses fail to expand longevity [20]. Lastly, alterations in mitochondrial functionality leading to premature aging do not cause increased ROS production [21,22]. In parallel, other studies have been conducted on the role of ROS as signaling molecules, demonstrating their importance in proliferation and survival pathways [23]. In conclusion, these discrepancies could be combined, considering ROS as signaling molecules essential to cell survival. Unfortunately, the excessive increase in ROS levels during aging overcomes the levels required for the proper signaling function, eventually magnifying, rather than ameliorating, the aging condition.

Beyond the fundamental role exerted by ROS signaling, age-related mitochondrial alterations have been associated with hormesis. A phenomenon characterized by the idea that a low dose of toxic stimulation triggers beneficial effects overcoming the restoration of the triggering damage. For instance, although severe damage to mitochondrial functionality is detrimental, mild respiratory impairments could increase lifespan due to a hormetic response [24]. Interestingly, metformin extends the lifespan in *C. elegans,* inducing a compensatory stress response by activating AMPK and Nrf2 [25].

### 2.2. mtDNA Mutations during Aging

Multiple mtDNA point mutations accumulate in different organs during aging in humans, monkeys, and rodents [26]. The first evidence that mtDNA mutations contribute to the aging process comes with the discovery that mtDNA mutations cause systemic diseases resulting in the same clinical manifestation related to the elderly. Indeed, mtDNA mutations affect the same tissues involved in aging, such as skeletal muscle, brain, and heart [26]. Subsequently, the generation of the “mitochondrial mutator mouse” has been excellent proof of the causative role exerted by mtDNA mutations during aging. Specifically, this mouse is a knock-in model expressing a mutated proofreading-deficient form of the nuclear-encoded mitochondrial DNA polymerase (Polg). This mutation causes a defect in the proofreading action of the enzyme, leading to the random accumulation of mtDNA mutations. The phenotype of this mouse is entirely normal during the first part of its life. However, it shows progressively premature aging behaviors such as alopecia, kyphosis, osteoporosis, anemia, and weight loss [22]. Furthermore, the mtDNA mutator mouse shows a decrease in the respiratory chain enzyme activities and, consequently, ATP production, mimicking what happens during the aging process. Interestingly, no evidence of enhanced ROS production was shown [21]. All these data suggest that the behaviors of the mtDNA mutator mouse are the consequence of mutations in protein-coding genes causing the alterations in the respiratory chain complexes, which, in turn, lead to premature aging.

### 2.3. PGC1-α Function Is Critical during Aging, Especially in Skeletal Muscle

Peroxisome proliferator-activated receptor-gamma coactivator-1α (PGC-1α) is considered the master regulator of mitochondrial biogenesis, acting as a transcriptional activator of several gene pathways controlling a variety of mitochondrial activities, including oxidative phosphorylation, fatty acid oxidation, scavenging activities, and mitochondrial dynamics [27,28]. In skeletal muscle, PGC-1α expression levels are decreased during aging [29]. Most importantly, enhanced PGC-1α levels prevent muscle wasting by decreasing apoptosis, autophagy, and proteasome degradation. These data suggest that PGC-1α could be a potential candidate for treatments to prevent or ameliorate age-related dysfunctions [30]. Finally, PGC-1α overexpression is sufficient to extend the lifespan and increase mitochondrial activities in flies [31].

It is worth mentioning that skeletal muscle represents the tissue where the link between aging and mitochondrial dysfunction has been deeply studied. In skeletal muscle, the aging process is accompanied by decreased activity of mitochondrial enzymes, a decline in the respiratory capacity, and increased ROS production. In addition, several studies have demonstrated a decrease in muscle mass and strength. Therefore, until now, the best intervention to counteract sarcopenia is physical exercise. In particular, endurance exercise benefits the elderly by stimulating mitochondrial biogenesis in different tissues [32].

## 3. Resveratrol

### 3.1. Chemistry, Safety, and Bioavailability of Resveratrol

Resveratrol is a natural phenol and a phytoalexin (from Greek aléxein = to safeguard); thus, it can protect plants from dangerous conditions such as excessive exposure to UV radiation and pathogens invasion [33]. In detail, the structure of resveratrol is characterized by two phenolic rings with a double bond; thus, it takes a planar structure, which provides hydrophobic behavior to the molecule. Moreover, due to the presence of three hydroxyl groups, resveratrol exerts a great antioxidant function and protects several cellular components from dangerous oxidation (Figure 2A) [34]. Indeed, as mentioned above, oxidative stress is strictly tuned in the cell and plays a critical role during aging. For this purpose, several studies have recently focused on polyphenols and other natural compounds with explicit antioxidant behavior, such as resveratrol.

The primary dietary sources of resveratrol are grapes (Vitis vinifera), peanuts, cocoa, and some berries such as strawberries and cranberries [35]. However, as a source of extraction for the production of nutritional supplements, Polygonum cuspidatum’s roots, a perennial plant native to East Asia, are preferentially used [36]. The interest in resveratrol began in the 1990s due to the so-called “French paradox”. Epidemiological studies conducted on the French population demonstrated decreased incidence of obesity and cardiovascular diseases even if its diet was rich in meat, cheese, and other components with a high content of cholesterol and saturated fat [37]. Firstly, the explanation was the increased consumption of red wine in the French population, which contains a modest amount of resveratrol (6.8 mg/L); however, further studies denied this theory by determining other dietary components’ beneficial effects on the cardiovascular system, such as fruit, vegetables, and olive oil, which contain a high number of polyphenols [38].

Both in vitro and in vivo experiments have demonstrated the positive effect of resveratrol on cellular homeostasis, particularly inflammatory reduction, antioxidant, and anti-aging effects. Resveratrol is considered a promising target to ameliorate several pathological conditions, such as chronic disorders and neurodegenerative diseases.

Regarding resveratrol’s safety, the European Food Safety Authority (EFSA) approved the use of this molecule as a food supplement, considering resveratrol safe and free from side effects for a daily dose of 150 mg/day [39]. In particular, resveratrol intestinal absorption is elevated due to the non-polar behavior of the molecule. Indeed, the small intestine absorbs 70% of the amount consumed. However, the enhanced hepatic metabolism reduces resveratrol bioavailability [40].

### 3.2. Mitochondrial Mechanisms of Action of Resveratrol

As mentioned above, resveratrol has multiple healthful behaviors, such as antioxidation, anti-inflammation, and the modulation of gut microbiota. Therefore, this molecule is considered a promising target to ameliorate several pathological conditions such as diabetes, cardiovascular pathologies, and neurodegenerative diseases. Mitochondria can be the target of its beneficial effect, given that resveratrol enhances mitochondrial function in diverse ways. Firstly, it induces the expression of genes involved in oxidative phosphorylation and mitochondrial biogenesis. In particular, mice treated with resveratrol show increased running capacity and OCR in muscle fibers. This phenomenon is due to the upregulation of oxphos genes and mitochondrial biogenesis. The mechanism of action is through SIRT1 activation, which, in turn, deacetylates PGC1α, thus activating it. This was confirmed by SIRT1 knockout cells treated with resveratrol, where no effect on enhanced mitochondrial functionality is visible [41]. Another study strengthened this molecular mechanism. Resveratrol-treated rats show increased aerobic performance due to the activation of the AMPK-SIRT1-PGC1α axis [42]. Furthermore, the combination of physical activity and resveratrol supplementation has been demonstrated to increase the expression of oxphos-related genes in a multifactorial accelerated mouse model used in aging research [43]. These results suggest that dietary resveratrol supplementation ameliorates mitochondrial functions in terms of the oxygen consumption rate and mitochondrial biogenesis. However, it should be noted that resveratrol inhibits F_0_F_1_-ATPase activity in a concentration-dependent manner, as shown in isolated mitochondria from rats [44].

Notably, the molecular mechanism involved in AMPK activation by resveratrol has remained an unsolved issue for many years. On the one hand, Vingtdeux and colleagues demonstrated that resveratrol increases cytosolic Ca^2+^ levels with the consequent activation of CaMKKβ, a kinase upstream of AMPK [45]. On the other hand, it has been proposed that the resveratrol-related effect on AMPK is due to its mild inhibition of F_0_F_1_-ATPase activity [44]. Park and coworkers provided convincing evidence in favor of the first hypothesis. Resveratrol increases cAMP levels due to its competitive inhibition of cAMP-degrading phosphodiesterases (PDEs), resulting in the canonical activation of the cAMP cascade, which involves the CamKKβ-AMPK pathway (Figure 3). Consequently, resveratrol increases NAD^+^ levels and SIRT1 activation. The proof of concept was performed using Rolipram, a PDE inhibitor, which reproduces the metabolic effect of resveratrol [46].

Beyond the fundamental role of resveratrol as an oxidative phosphorylation enhancer, it is known as one of the principal antioxidant compounds, as demonstrated in different cell types, including microglia, cancer cells, and endothelial cells. Resveratrol regulates the cellular redox balance using two distinct mechanisms. It acts directly as a potent scavenger of hydrogen peroxide, superoxide anion, and hydroxyl radicals. Moreover, it can work indirectly. On the one hand, it stimulates endogenous systems that act against ROS. It has been demonstrated that resveratrol activates the Nrf2 pathway in the primary culture of human keratinocytes [47]. Nrf2 is a basic leucine zipper transcription factor that controls the expression of crucial antioxidant cellular mechanisms. Among the Nrf2-activated genes, there are genes involved in glutathione metabolism. Not only human keratinocytes, but also arterial endothelial cells treated with resveratrol show enhanced Nrf2 transcriptional activity [48]. On the other hand, resveratrol decreases ROS production by SIRT1 activation. SIRT1 overexpression mimics the resveratrol effect, whereas SIRT1 silencing attenuates the resveratrol-induced antioxidant activity [49].

### 3.3. Beneficial Effects of Resveratrol during Aging and Age-Associated Diseases

The aged population is increasing. As a result, in this decade, we will experience duplication in the percentage of people aged 60 years and older [50]. In parallel, the risk of susceptibility to different age-related diseases is inflating, including cardiovascular diseases, diabetes, sarcopenia, and neurodegenerative diseases. Thus, all strategies targeting age-related decline are likely to significantly impact current and future societies.

Multiple studies have demonstrated the positive antioxidant effect exerted by resveratrol in vitro and in vivo models of aging [51]. This part will focus on how resveratrol treatment can ameliorate some age-related aspects acting on mitochondrial functionality (Table 1).

Sarcopenia is a typical age-related condition characterized by progressive muscle mass loss and function loss. Resveratrol could counteract this skeletal muscle decline by ameliorating mitochondrial homeostasis. In particular, the combination of short-term caloric restriction and resveratrol treatment shows mild protective effects in terms of muscle mass and PGC1α levels in the glycolytic muscles of aged rodents [52]. Another study demonstrated that the combination of exercise training and resveratrol supplementation is able to improve the muscle performance of aged mice. The increased antioxidant enzyme activities and the decreased muscle lipid peroxidation are accompanied by enhanced PGC1α expression, suggesting that resveratrol and exercise training could sustain mitochondrial biogenesis and function, which, in turn, ameliorate muscle activity [53]. Furthermore, resveratrol not only acts on mitochondrial metabolism to improve muscle function during aging, but it also modulates the sarcomere structure by activating the AMPK pathway [54]. In summary, resveratrol alone has a mild effect against sarcopenia; however, in combination with exercise or caloric restriction, it is able to counteract muscle mass loss.

Regarding human studies, resveratrol supplementation triggers the activation of the AMPK-SIRT1-PGC1α axis in the skeletal muscle of obese subjects [55]. However, neither resveratrol-induced improvement of metabolic function nor resveratrol-induced protection against inflammatory state have been observed in aged men subjected to exercise training [56]. These misleading results come from highly variable clinical trials. Notably, most of these studies have a limited sample size. Thus, more effort is needed to clarify how much is the beneficial contribution of resveratrol supplementation on health [57].

Besides skeletal muscle, cardiovascular functionality is also involved in age-related decline. In particular, cardiovascular diseases are the leading cause of death worldwide. Several in vitro studies conducted on endothelial cells demonstrated that resveratrol treatment is sufficient to reduce oxidative stress [58,59,60]. Doxorubicin-induced impairment of cardiac systolic function in aged animals is suppressed by resveratrol treatment due to the activation of the SIRT1 pathway [61]. Moreover, resveratrol protects against arterial aging by regulating the renin–angiotensin axis [62]. Finally, Aldh2, a mitochondrial key player in the regulation of cardiac homeostasis, accentuates age-related mitochondrial alterations. Importantly, these dysfunctions could be mitigated by resveratrol [63].

Regarding human studies, most were conducted on patients affected by cardiovascular diseases instead of aged subjects. Nevertheless, several works reported the positive impact of resveratrol on these patients (for a comprehensive review, see [64]). Notably, oral resveratrol supplementation improves glycemic control in terms of systemic blood pressure and total cholesterol levels in patients with type 2 diabetes [65]. Moreover, resveratrol decreases total cholesterol levels and inflammatory markers in patients with angina [66]. However, many subjects were on multiple drugs; thus, this fact should be considered in interpreting the data. In summary, resveratrol has beneficial effects in ameliorating some age-related features, especially in subjects with metabolic-related pathologies.

Aging is also associated with neurodegenerative diseases, a group of progressive pathologies accompanied by inflammatory status. Experimental evidence has demonstrated that resveratrol protects against neurodegeneration and preserves cognitive functions [67]. In particular, resveratrol induces neuronal differentiation in adult hippocampal precursor cells [68]. Moreover, an intraventricular resveratrol injection improves long-term memory formation. Notably, these effects are absent in SIRT1 mutant mice [69]. In addition, it enhances neurogenesis and micro vascularization [70].

To date, there is limited and discordant information about resveratrol’s role in reducing brain inflammation in humans. For instance, a study on healthy patients demonstrated that resveratrol increases cerebral blood flow without affecting cognitive function [71]. On the other hand, another study shows that resveratrol supplementation improves memory performance and hippocampal function [72]. Moreover, several clinical trials are currently in progress focused on resveratrol effects in neurodegenerative diseases. In conclusion, it is well established that resveratrol plays a positive role in vitro and in animal models; however, more effort should be made to unveil its effect on human cognitive function.

Type 2 diabetes mellitus (T2DM) represents a severe problem in the coming years. One in three American adults will develop diabetes by 2045, according to the predictions made by the American diabetes association [73]. An effective treatment is also needed because of the association of diabetes with other pathological conditions such as heart dysfunction, neuropathy, and retinopathy.

As already mentioned, AMPK plays a leading role in glucose-stimulated insulin secretion; indeed, the best treatment, until now, to counteract T2DM is metformin, a well-known AMPK activator. In addition, AMPK is involved in metabolic syndrome as a hypothalamic nutrient sensor. Therefore, since resveratrol causes the activation of AMPK, it could be a promising agent against obesity and diabetes.

Studies on primates fed with a high-fat/high-glucose diet report that daily resveratrol supplementation has beneficial effects on glycemia, β cell functionality, and inflammatory state [74,75]. Controversial results are present in human studies where the efficacy of resveratrol treatment seems to depend on the metabolic status of subjects.

Diabetic patients, treated with resveratrol as an adjunct to canonical pharmacological therapy, show improvement in some cardiometabolic markers but not glycemia, insulin levels, and lipid metabolism [76]. However, another meta-analysis demonstrated that resveratrol supplementation significantly enhances glucose control and insulin sensitivity only in T2DM patients and not in non-diabetic subjects [77]. To conclude, resveratrol could be used as an adjuvant in combination with classical T2DM pharmacological treatments.

Finally, gestational diabetes, a condition diagnosed for the first time during pregnancy, is associated with high blood glucose and could negatively affect fetal development. Recent studies on animal models of gestational diabetes demonstrated that resveratrol supplementation during pregnancy restores normal glycemia and insulin secretion [78,79]. Even if these promising data could pave the way for resveratrol treatment to counteract gestational diabetes, more effort should be made.

**Table 1 nutrients-14-04326-t001:** Studies related to resveratrol treatment in different physiopathological conditions.

Effects of Resveratrol On	Beneficial Effect On	Studies On	Ref.
skeletal muscle	muscle mass	rodents	[52]
	muscle performance	rodents	[53]
	sarcomere structure	rodents	[54]
	activation of AMPK-SIRT1 pathway	humans	[55]
	no difference in the inflammation state	humans	[56]
cardiovascular impairment	systolic function	rodents	[61]
	renin-angiotensin axis	rodents	[62]
	aged-related mitochondrial dysfunctions	rodents	[63]
	glycemic control	T2DM patients	[65]
	decreased cholesterol levels	patients with angina	[66]
neurodegenerative diseases	increased long-term memory formation	rodents	[69]
	increased neurogenesis and vascularization	rodents	[70]
	increased cerebellar blood flow	humans	[71]
	increased memory performance	humans	[72]
diabetes	enhanced cardiometabolic markers without affecting glycemia	humans	[76]
	increased glucose control	humans	[77]
	normal glycemia restoration	gestational diabetes	[78,79]

## 4. CoenzymeQ10

### 4.1. Chemistry of CoQ10

Coenzyme Q (CoQ or ubiquinone) is an essential lipid in all cell membranes but is especially abundant in mitochondria. It is formed by a polar head consisting of a quinone group bound to a polyisoprenoid tail of variable length depending on the species (six units for yeast, nine for mice, and ten for humans) [80,81] (Figure 2B). It is fundamental for cell function by acting as an electron shuttle in the mitochondrial respiratory chain. It shuttles electrons from complexes I and II to complex III. Remarkably, it is the only fully hydrophobic component of the ETC. Moreover, it acts as a cofactor for multiple dehydrogenases, a modulator of the mPTP (mitochondrial permeability transition pore), and an antioxidant molecule able to prevent oxidative damage to membranes [80,82]. The antioxidant features of the reduced form of CoQ_10_ were first described during the 1960s in submitochondrial particles [83,84]. Since CoQH_2_ is oxidized during lipid peroxidation, its regeneration is mandatory to ensurethe antioxidant function. In fact, mitochondrial CoQH_2_ is mainly reduced by complex III. However, the reduced form of CoQ can also be regenerated at additional cell locations. For instance, different cytosolic enzymes can efficiently reduce CoQ_10_, including cytoplasmic NADPH-dependent CoQ reductase, lipoamide dehydrogenase, and thioredoxin reductase [85,86,87].

The highly conserved CoQ_10_ biosynthesis pathway is the result of its pleiotropic role [88]. CoQ is mainly produced inside the mitochondria and then distributed to cell membranes. In addition, it is present in several foods; thus, it can be obtained from the diet. Presently, the biosynthesis pathway is still under investigation. Notably, many of the genes involved in this pathway were discovered due to studies conducted in human patients with CoQ_10_-deficient syndrome.

### 4.2. CoQ_10_ as a Key Factor in Controlling Cellular Homeostasis

This section will focus on the CoQ_10_ regulation of the cellular redox balance.

Since the critical function of CoQ is to function as an electron carrier in the ETC, CoQ redox reactions are linked directly or indirectly to oxidative phosphorylation. Indeed, most CoQ redox reactions come from NADH-dependent complex I and FADH_2_-dependent complex II. In addition, CoQ could also be reduced by different oxidoreductases [89]. Hereafter, some of them are listed: Glycerol-3-phosphate dehydrogenase (G3PDH), which integrates oxidative phosphorylation, glycolysis, and FA metabolism [90]; electron-transport flavoprotein dehydrogenase (ETFDH), which plays a role in FA and branched amino acids oxidation [91]; sulfide-quinone oxidoreductase involved in sulfide detoxification [92]; and proline dehydrogenase, which participates in glyoxylate metabolism [93].

CoQ is also important as a structural element to maintain complex I and III [94,95]. Interestingly, CoQ supplementation in CoQ-deficient yeast can restore complex III assembly, suggesting that it plays an essential role in maintaining this complex [96]. Moreover, it participates in the complex I assembly. The reduced-to-oxidized CoQ ratio allows the reconfiguration of the ETC ultrastructure when a shift from glucose to FA occurs. In particular, an accumulation of ubiquinol in the ETC destabilizes complex I [97].

CoQ is also a component of respirasome [98], thus, taking part in supercomplex assembly. It has been proposed that Opa1, a key player in regulating OMM fusion, regulates complex IV activity in a CoQ-dependent fashion [99]. Intriguingly, supercomplexes composed of complexes I, II, and IV are associated with a specific CoQ poll, whereas free CoQ is committed to complex II [92].

Extramitochondrial ubiquinol acts as an antioxidant in different cellular systems. Together with vitamin E, CoQ scavenges the lipid peroxyl radicals, thus avoiding the initiation of lipid peroxidation [100]. To work as an antioxidant agent, ubiquinol requires the participation of different ubiquinone reductases, which convert the oxidized ubiquinone into ubiquinol. In this way, the latter can be reused. Thus, different cytosolic oxidoreductases have been proposed for this function. In particular, the family of pyridine nucleotide oxidoreductases takes care of extramitochondrial ubiquinone reduction [100]. Glutathione reductase and lipoamide dehydrogenase are members of this family.

Eventually, CoQ exerts an antioxidant role by preventing oxidative damage in the low-density lipoproteins (LDLs), which is essential for the protection against cardiovascular diseases [101,102]. Takahashi and colleagues show that an LDL-associated CoQ-reducing mechanism exists at the external surface of the plasma membrane in HepG2 cells [103]. However, other studies are needed to elucidate the molecular mechanisms that maintain the CoQ redox cycle in the extracellular environment.

### 4.3. Beneficial Effects of CoQ_10_ during Aging and Age-Associated Diseases

A reduction in CoQ_10_ biosynthesis has been linked to aging and aging-related diseases, suggesting that its decrease may contribute to the decline in cellular function typical of the aging process (Figure 4). CoQ_10_ increases over the lifespan of up to 18 months, then decreases significantly [104]. However, controversial results arise when researchers tried to correlate CoQ levels and lifespan. Indeed, although CoQ_7_ heterozygous worms and mice have the same CoQ protein levels as controls, they exhibit increased lifespans. This result suggests that CoQ per se is not responsible for extended longevity [105]. On the other hand, Tian and colleagues demonstrated that dietary CoQ supplementation has beneficial effects in a senescence-accelerated mouse model in terms of increased mitochondrial biogenesis. In particular, CoQ decreases sirtuins expression resulting in the activation of PCG1α [106].

Aging per se is not the only variable affecting CoQ levels; exercise also plays a crucial role. Physical activity increases the CoQ_10_ levels during aging. Elderly subjects with higher muscular strength have increased CoQ levels in the bloodstream, accompanied by low levels of cholesterol and lipid peroxidation [107]. In addition, a low CoQ_10_H_2_:CoQ_10_ ratio could be used as an indicator of enhanced risk of sarcopenia [108]. Finally, 4-year CoQ_10_ supplementation in older people provides increased physical performance [109] (Table 2). These data are promising, even if more human clinical trials should be performed to postulate that CoQ could be used to maintain muscle performance during aging.

A few clinical trials have been conducted on patients with different neurodegenerative pathologies. Among them, Parkinson’s disease (PD) shows some positive results, even if there are controversial studies. For example, a clinical trial demonstrated that different doses of CoQ_10_, up to 1200 mg/day, are well tolerated by Parkinson’s disease patients, who show a reduced development of disability [110]. On the contrary, metadata analyses indicate no difference in motor symptoms between PD patients and controls [111,112].

As mentioned above, cardiovascular diseases are a global health issue. Since ROS production is crucial during heart failure, antioxidant therapies have been considered to prevent this pathology. Systematic reviews and meta-analyses have been conducted to evaluate the efficacy of CoQ supplementation in preventing cardiovascular diseases. In detail, Flowers and colleagues demonstrated that CoQ treatment is able to reduce systolic blood pressure without affecting other factors, such as diastolic blood pressure or cholesterol levels [113]. In addition, in patients undergoing cardiac surgery, CoQ treatment prevents some complications, such as arrhythmias [114]. In line with these results, a short CoQ treatment increases the left ventricular ejection fraction of patients with congestive heart failure [115]. Interestingly, 2-year CoQ treatment in patients with chronic heart failure reduces major cardiovascular events [116]. All these data indicate a positive effect of CoQ supplementation in counteracting heart failure and related complications.

Due to its antioxidant capacity, CoQ was evaluated not only to prevent cardiovascular diseases but also to treat diabetes and metabolic syndrome. Until now, there have been no sufficient data to determine the effect of CoQ in this group of diseases. Hereafter, we report a few recent clinical trials or meta-analyses. Two systemic reviews have reported incoherent results that analyze the effects of CoQ treatment in diabetes patients. It has been demonstrated that CoQ supplementation in diabetic patients slightly reduces fasting blood glucose [117]. However, another research group, which analyzed more than seven trials, reported no differences in glycemic control as a result of CoQ [118]. Surprisingly, the same study discovered that CoQ may reduce triglyceride levels. To corroborate these data, treatment with a blend of different natural compounds, including CoQ, astaxanthin, and others, significantly reduced triglycerides and cholesterol levels. Unfortunately, this reduction was not due to the CoQ effect per se [119]. Finally, it has been demonstrated that old people receiving a CoQ and selenium treatment for 4 years show increased IGF-1 levels compared to placebo-treated patients [120].

CoQ treatment is also used to counteract the side effects of statin therapy. Indeed, statins are associated with myopathy caused by the inhibition of 3-hydroxy-3-methylglutaryl-CoA reductase, which is a key player in CoQ biosynthesis [121]. A small group of patients with statin-associated myopathy has been treated with CoQ supplementation showing decreased muscle pain and fatigue [122]. This result is promising; however, it was conducted with a small number of patients and is not sufficient to state whether CoQ supplementation attenuates the side effects of chronic statin treatment.

**Table 2 nutrients-14-04326-t002:** Studies related to CoQ_10_ supplementation in different physiopathological conditions.

Effect of Coenzyme Q10 On	Beneficial Effect On	Ref.
elderly subjects	exercise performance	[109]
Parkinson’s disease patients	decreased development of disability	[116]
	no effect on motor symptoms	[105,113]
cardiovascular impairment	decreased systemic blood pressure	[107]
	preventing arrhythmias in cardiac surgery-subjected patients	[114]
	increased left ventricular ejection fraction	[115]
	decreased cardiovascular events in patients with chronic heart failure	[116]
diabetes	decreased blood glucose	[117]
	no differences in glycemic control	[118]
counteracting statins side-effects	decreased muscle impairment	[122]

### 4.4. CoQ Deficiency Syndrome

In this section, we will briefly describe the diseases associated with a reduced concentration of CoQ inside the cell; however, for a comprehensive review, see [123]. The first case of CoQ deficiency was documented in 1989. Ogasahara and colleagues reported two sisters who displayed muscle weakness and reduced respiratory complexes’ activity [124]. Until now, CoQ deficiency has been described as a group of heterogeneous mitochondrial diseases characterized by decreased intracellular CoQ levels.

Due to the crucial role exerted by CoQ in the mitochondria, its reduction leads to the impairment of oxidative phosphorylation with the final decrease in ATP production [97]. Moreover, other processes in which CoQ is involved could contribute to the disease progression.

CoQ deficiencies are divided into two different classes: primary and secondary deficiencies. CoQ primary deficiencies are autosomal recessive pathologies caused by mutations in genes involved in the CoQ biosynthesis pathway. It is a very heterogeneous group of diseases with a broad spectrum of clinical symptoms. For instance, the same mutation can lead to diverse clinical outcomes. In some patients, only a single organ is affected, while other subjects carrying the same mutation display a multiorgan disorder [125].

We refer to secondary CoQ deficiencies when the CoQ reduction is a secondary event due to the impairment in a process not involving CoQ biosynthesis. This group of pathologies can be classified based on the origin of the CoQ deficiency, i.e., due to oxidative phosphorylation alterations, impairment of cholesterol biosynthesis, or defects in mitochondrial homeostasis [126].

## 5. Challenges and Conclusions

Several in vitro studies corroborated by animal research demonstrate the beneficial effect of resveratrol treatment. However, discrepancies arise with human clinical trials for different reasons. Firstly, most of the studies have been carried out using small sample sizes. Secondly, a wide range of dosage levels was used, making it difficult to compare different studies. Thus, more effort should be made to put together data produced by clinical studies with in vitro research analyses to find a consensus.

In addition, different aspects need further clarification. First, resveratrol’s bioavailability is mainly due to rapid metabolic transformation. Indeed, new formulations were tested in recent years to extend it [127]. Another possible approach to overcome this problem is the development of prodrugs (for a comprehensive review, see [128]). Moreover, additional evidence is needed to clarify the beneficial effects of resveratrol in combination with other supplements. Finally, it is worth mentioning that high doses of resveratrol have severe side effects [129]. Thus, long-term studies are needed to determine the effect of resveratrol supplementation and its interaction with other drugs on human health.

CoQ is fundamental in maintaining cellular homeostasis, being a crucial component of the ETC and an antioxidant agent [82]. Thus, it is unsurprising that a decreased CoQ level affects various cellular processes. In addition, CoQ deficiency is associated with different age-related conditions [130].

Several in vitro studies suggested that CoQ supplementation could enhance mitochondrial functionality and exert tissue antioxidant protection. However, since meta-analyses showed contrasting results due to the small sample size, different doses, and short follow-up period, further work is needed to evaluate its beneficial effect on alleviating age-related alterations.

In general, caution is needed when interpreting human clinical trials and meta-analyses results. The association of diverse factors such as differences in the experimental design, number of patients engaged, treatment and dose, and follow-up period duration should be considered. Presently, we can consider resveratrol and CoQ_10_ two coadjuvant agents able to ameliorate the aging condition and some age-related diseases.

## Figures and Tables

**Figure 1 nutrients-14-04326-f001:**
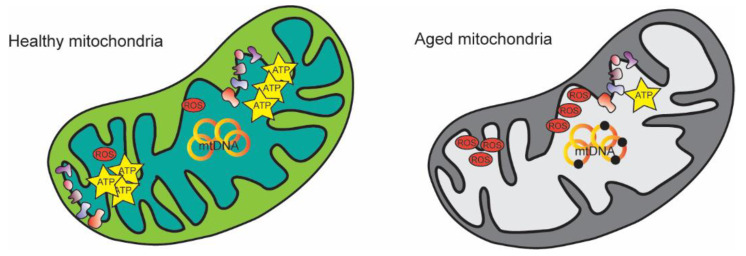
Mitochondrial changes during aging. Mitochondria display morphological and functional remodeling including abnormal cristae, decreased ATP production, increased mtDNA mutations, and alteration in ROS production.

**Figure 2 nutrients-14-04326-f002:**
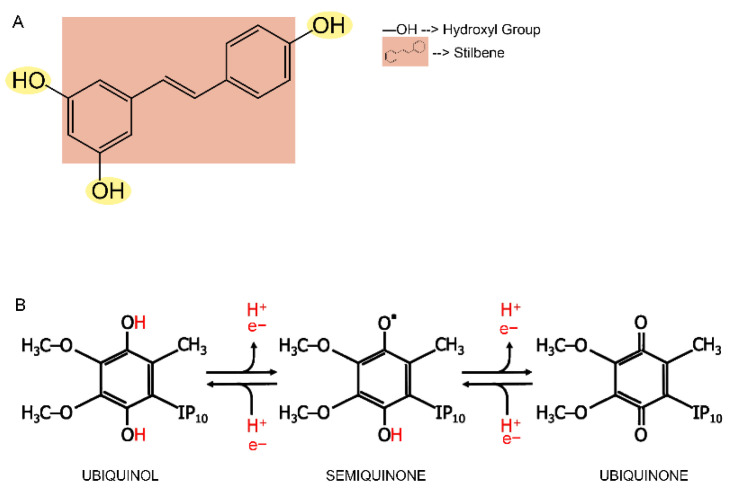
Chemical structures of resveratrol (RSV) and Coenzyme Q_10_ (CoQ_10_). (**A**) Resveratrol is a stilbene structure with three hydroxyl groups. (**B**) The CoQ_10_’s complete reduction needs two protons and two electrons. It is a two-step reaction with the formation of an intermediate called semiquinone.

**Figure 3 nutrients-14-04326-f003:**
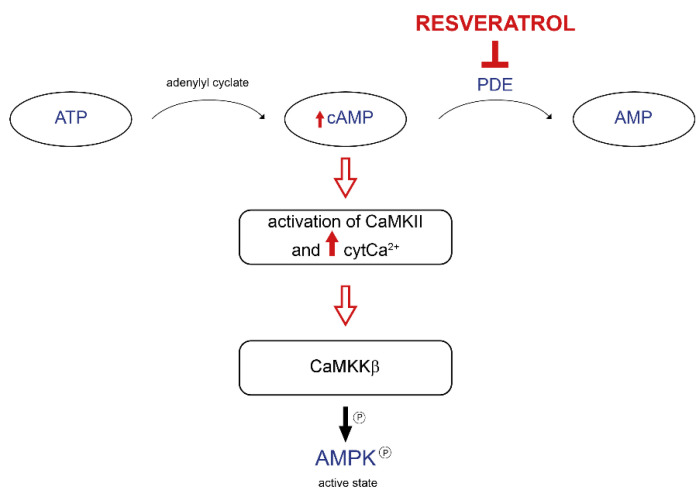
Resveratrol is involved in the signaling pathway, which leads to AMPK activation. Resveratrol inhibits phosphodiesterases (PDE), thus increasing intracellular cAMP concentration. cAMP elevation leads to the activation of CaMKII with the consequent increase in cytosolic Ca^2+^ concentration. Ca^2+^ activates CaMKKβ, one of the upstream kinases of AMPK.

**Figure 4 nutrients-14-04326-f004:**
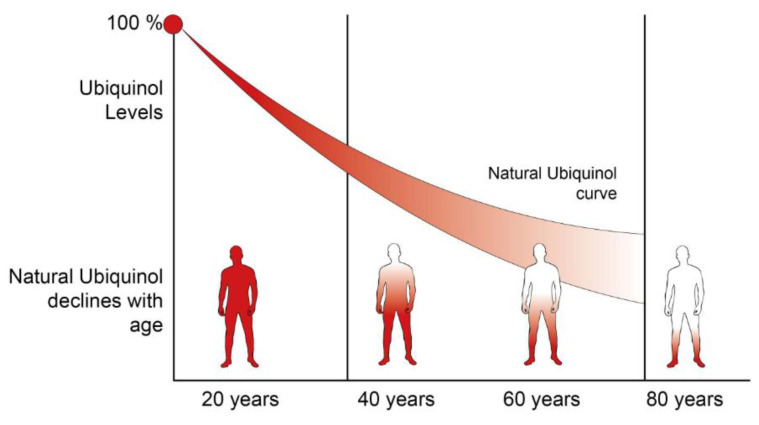
CoQ_10_ concentration decreases with aging. During adulthood, people start losing ubiquinol.

## Data Availability

Not applicable.

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
