# Peer review of "CoQ10 and Resveratrol Effects to Ameliorate Aged-Related Mitochondrial Dysfunctions"

_nutrients, 2022, doi:10.3390/nu14204326_

Round 1
Reviewer 1 Report
In this manuscript authors discussed about recent clinical trials and meta-analyses based on resveratrol and CoQ supplementation during elderly focusing on mitochondrial functionality. The manuscript reads well and I have following suggestions.
1. Introduction:
Some of the abbreviations are not explained well (Eg: Line 50- What is CoQ10 stands for?).
The sentences were written without supported literature cited. Only 2 references were cited for the whole introduction. I suggest to include more references to support the sentences.
2. Mitochondrial alterations during aging
This section could be sub sectioned for clarity with more information as the manuscript title is being mostly elaborated on this section. I suggest to include a figure for the explanation as well. Authors cited only one reference from line 69-81.
3. Resveratrol
This section is well explained.
4. CoenzymeQ10
References are not cited in the text.
Eg: Line 342,349, 358-Add the reference.
Section 5 briefly describes overall challenges and conclusions. However, citing references are not consistent. Eg: first paragraph does not include the reference.
Overall, I suggest citing references and adding some figures for more clarity as I discussed above.
Author Response
In this manuscript authors discussed about recent clinical trials and meta-analyses based on resveratrol and CoQ supplementation during elderly focusing on mitochondrial functionality. The manuscript reads well and I have following suggestions.
- Introduction:
Some of the abbreviations are not explained well (Eg: Line 50- What is CoQ10 stands for?).
We thank the reviewer for this comment. In the revised version of the manuscript we used the extended name. We started to use the abbreviation in the Coenzyme Q10-related section.
The sentences were written without supported literature cited. Only 2 references were cited for the whole introduction. I suggest to include more references to support the sentences.
We agree with the reviewer about the lack of some critical references. We added the following references in the new version of the manuscript:
[1] H. F. Deluca and G. W. Engstrom, “Calcium uptake by rat kidney mitochondria.,” Proc Natl Acad Sci U S A, vol. 47, no. 11, pp. 1744–50, Nov. 1961, Accessed: Oct. 14, 2016. [Online]. Available: http://www.ncbi.nlm.nih.gov/pubmed/13885269
[2] F. D. Vasington and J. v Murphy, “Ca ion uptake by rat kidney mitochondria and its dependence on respiration and phosphorylation.,” J Biol Chem, vol. 237, pp. 2670–7, Aug. 1962, Accessed: Oct. 14, 2016. [Online]. Available: http://www.ncbi.nlm.nih.gov/pubmed/13925019
[3] P. Bernardi, F. di Lisa, F. Fogolari, and G. Lippe, “From ATP to PTP and Back: A Dual Function for the Mito-chondrial ATP Synthase,” Circ Res, vol. 116, no. 11, pp. 1850–1862, May 2015, doi: 10.1161/CIRCRESAHA.115.306557.
[6] T. A. F. Aguilar, B. C. Hernández Navarro, and J. A. M. Pérez, “Endogenous Antioxidants: A Review of their Role in Oxidative Stress,” A Master Regulator of Oxidative Stress - The Transcription Factor Nrf2, Dec. 2016, doi: 10.5772/65715.
[7] N. Sun, R. J. Youle, and T. Finkel, “The Mitochondrial Basis of Aging,” Mol Cell, vol. 61, no. 5, p. 654, Mar. 2016, doi: 10.1016/J.MOLCEL.2016.01.028.
[8] T. Lima, T. Y. Li, A. Mottis, and J. Auwerx, “Pleiotropic effects of mitochondria in aging,” Nature Aging 2022 2:3, vol. 2, no. 3, pp. 199–213, Mar. 2022, doi: 10.1038/s43587-022-00191-2.
[9] T. Miyazawa, C. Abe, G. C. Burdeos, A. Matsumoto, and M. Toda, “Food Antioxidants and Aging: Theory, Current Evidence and Perspectives,” Nutraceuticals 2022 NUTRACEUTICALS2030014., Vol. 2, Pages 181-204, vol. 2, no. 3, pp. 181–204, Aug. 2022, doi: 10.3390/
- Mitochondrial alterations during aging
This section could be sub sectioned for clarity with more information as the manuscript title is being mostly elaborated on this section. I suggest to include a figure for the explanation as well. Authors cited only one reference from line 69-81.
We thank the reviewer for the critical points raised in this section. We sub-sectioned the text introducing other data and changing the order of some other info. In addition, we added a figure to make the concept of aged mitochondria clearer. Eventually, we add key references in the first two paragraphs.
- Resveratrol
This section is well explained.
We thank the reviewer for this comment.
- CoenzymeQ10
References are not cited in the text.
Eg: Line 342,349, 358-Add the reference.
We thank the reviewer for this comment. We arranged in the correct way the references.
Section 5 briefly describes overall challenges and conclusions. However, citing references are not consistent. Eg: first paragraph does not include the reference.
We thank the reviewer for this comment. The aim of the conclusion paragraph is to sum up all the works cited in the previous sections and, most importantly, to make a more general and critical comment based on the data collected to write this review. Thus, since the conclusions are made by the authors themselves, we think that references could be inappropriate. However, we added 2 more references where appropriate.
Overall, I suggest citing references and adding some figures for more clarity as I discussed above.
Reviewer 2 Report
General comments:
The manuscript titled “CoQ10 and resveratrol effects to ameliorate aged-related mitochondrial dysfunctions” by Gherardi et al. provides a compendium of the latest advances on the potential role of the mitochondrial electron carrier CoQ10 and the reactive oxygen and nitrogen species scavenger resveratrol in mitigating age-related mitochondrial dysfunction. The premise of the work is that “any treatment…to improve and sustain mitochondrial functionality is a good candidate to counteract mitochondrial-associated dysfunction” during aging. In this review, the authors provided insights into the use of these compounds in recent clinical trials and metanalysis as it relates to mitochondrial changes during aging. The authors endeavored to provide a balance coverage of the matter, highlighting controversies where possible. In addition, the authors presented potential cautionary factors to be considered when considering experimental and clinical cases, and dose and duration in administering the supplements. They are commended for this diligent and masterful effort. However, my enthusiasm for the work was significantly diminished by the low quality of the presentation. It was disappointing to see the quality of writing in this manuscript to be below what one would expect for a reputable journal. The entire manuscript was challenging and laborious to read. Unfortunately, these negative attributes overshadowed what I initially thought would be an interesting topic to read. There are sentences that lack punctuations in the right place or contain misplaced punctuations, and spelling errors or use of wrong words. Some sentences are vague and awkward. This makes for a tough reading. The authors are highly recommended to seek professional help before resubmitting a revision, i.e., if they are accorded the opportunity. Any future resubmission should be accompanied by a certificate of authentication, showing proof of professional editing assistance. There is also lack of consistency in the use of tenses, past and present, and this also contributes towards the complicated nature of the manuscript. It will take an inordinate amount of time to pinpoint all the flaws and grammatically challenging phrases in this manuscript. In this critique, only highlights of some glaring ones will be noted. The onus for good presentation is on the authors. They should do a thorough job polishing and fine-tuning their paper before submission. Lastly, the authors must avoid using this phrase “thanks to the…” repeatedly.
Specific comments
Abstract:
Line 19 “…function during the elderly…” Revise to read ““…function during the aging process…” There several areas in the manuscript where similar phrases are used. These should all be corrected. Lines 19-20 “In the last years interest is growth on natural compounds able to modulate mitochondrial function…” is an awkward sentence and must be revised. Lines 21-23 “…we will discuss about recent clinical trials and meta-analyses based on resveratrol and CoQ supplementation during elderly…” is another sentence that must be revised for clarity of meaning. There are other minor syntax errors that the authors must address.
1. Introduction
Lines 28-30 “…in that case, glycolytic products could be oxidized in a mitochondrial-independent way in the cytosol whit a less efficient anaerobic respiration” This sentence could be simplified for better understanding. Lines 42-44 “The ETC is the major reactive oxygen species (ROS) producer, indeed, although oxidative phosphorylation is an efficient process, a small amount of electrons leaks from the ETC and interacts with oxygen forming ROS…” Revise to read, “The ETC is the major source for reactive oxygen species (ROS) production; indeed, although oxidative phosphorylation is an efficient process, a small amount of electrons leaks from the ETC and interacts with molecular oxygen to produce ROS…” Line 45, “aging” should be added at the end of the phrase “…”free radical theory…” Lines 51-54 “Although mitochondria play a central role in the control of energy metabolism, today we can no longer limiting our view only on their energetics behaviors, but rather mitochondria are platforms for many intracellular signals interfering in many different physiological and pathological settings.” This entire sentence is complex and reads awkward. It must be revised. In this section, other issues, such as poor punctuations, abound. The entire section must undergo thorough review.
2. Mitochondrial alterations during aging
Line 69 “Several studies show that aging is accompanied by a drop of mitochondrial function…” Revise to read, “Several studies show that aging is associated with a decrease in mitochondrial function…” This sentence, lines 91-92, “In conclusion, these discrepancies could be combine together considering ROS a necessary survival signal induced by different stress conditions with the scope of ameliorate the progressive…” is grammatically flawed and must be revised. This sentence, lines 97-99, “On the other hand, in elderly ROS production increases overcoming the levels required for the proper signaling function, eventually magnifying, instead of being beneficial, the aging condition…” is tough to decipher because of the poor structure of the sentence. Line 111 “…progressively acquires different features of premature aging such as alopecia…” reads awkward. Consider revising line 128 “Mitochondrial alterations during aging are also linked to mitohormesis, meant as the…” is a grammatically poor sentence and must be revised. In the sentence, lines 131-133, what does “…different compounds such as metformin and resveratrol are mild mitochondrial poisons…mean, even though they can activate AMPK? Lines 139-141 “Until now, the best intervention…stimulating mitochondrial biogenesis…” should be revised. In addition to these comments, there are other syntax errors and bad punctuations that must be corrected. This will improve the readability and understanding of the sentence.
3. Resveratrol
3.1. Chemistry, safety and bioavailability of resveratrol
Lines 146-147 “In detail, resveratrol is characterized by two phenolic rings with a double bound, thus resveratrol…” Revise to read, “The structure of resveratrol is characterized by two phenolic rings with double bonds, thus resveratrol…” Line 151, what do the authors mean by “…the oxidative stress should be strictly tuned in the cell…”? Line 185 “In this contest…” It is unclear what the authors are trying to say here. “Contest” does not seem to be the right word here. Please explain. In the same sentence, what does the phrase “…mitochondria are at the center of its beneficial effect…” mean? Line 200 “…it should be taken into account that resveratrol is able to inhibit the F0F1-ATPase activity in a concentration-dependent manner…” Simplify to read, “…it should be noted that resveratrol inhibits the F0F1-ATPase activity in a concentration-dependent manner…” The paragraph, lines 203-213, needs a significant revision, including use of proper punctuations and the proper use of the article “the”. Similarly, the entire paragraph, lines 219-236, must be revised. It has awkward phrases, poor syntax and improper punctuations. These issues make it challenging for one to read and understand the text. For example, what does the phrase “…resveratrol could have an indirect antioxidant effect based on its capacity to modulate the expression of proteins induced in during scavenger activity…” mean?
3.3. Beneficial effects of resveratrol during aging and aged-associated diseases
Just like the previous sections, this entire section also requires careful scrutiny of the language structure. Too many tough sentences/phrases and unclear meanings. Line 243 “Since the first evidence that resveratrol could act mimicking some anti-aged effects of dietary restriction, multiple studies have been accumulated demonstrating the positive…” What does this sentence mean? The sentence, “Different studies demonstrated that resveratrol could counteracts this situation impinging on mitochondrial homeostasis…” must be revised and simplified. Line 260, what does this “…it also modulates the sarcomere structure impinging on AMPK pathway…” mean? Line 264 “Conflicting results have emerged in human studies…” What are the “conflicting results”? Line 271 “Besides skeletal muscle also cardiovascular functionality, in terms of endothelial dysfunction and alteration in the redox balance, is involved in age-related…” Simplify. The paragraph, lines 292-308 contain unusual sentences, poor syntax and improper use of punctuations. Difficult to understand the message being conveyed. Lines 317-318 “These assumption indicate resveratrol as a promising agent against obesity and diabetes…” is a flawed sentence. Lines 321-322 “Concerning human studies, even if controversial data are present, it is clear that the efficacy of resveratrol treatment depends on the metabolic status…” What does this mean? Revise. Lines 324-326 “…as an adjunct to the canonical pharmacological therapy show that this molecule is able to improve some cardiometabolic markers but not glycemia, insulin levels and lipid metabolism…” This is a complicated sentence to decipher, and meaning is unclear. In the phrase, “Finally, gestational diabetes, not less important than T2DM…”, why is this information comparing the importance of the two types of diabetes necessary here to mention here?
4. CoenzymeQ10
4.1. Chemistry of CoQ10
The entire paragraph, lines 339-356, must undergo extensive review for proper grammar and syntax use. What is meant by “by a polar head bound”? “It is worth to mention that the reduced form of CoQ can be regenerated at different cell location, not only into the mitochondria.” A reference/s must be cited here. Lines 357-363 should include some citations, to verify some of the claims made.
4.2. CoQ10 as a key factor to control cellular homeostasis
This entire section must undergo stringent reevaluation for syntax, grammar and presentation.
Line 365 “In this section we will focused…” This is an astounding and glaring grammatical error.
4.3. Beneficial effects of CoQ10 during aging and age-associated diseases
This section should be revised to improve its readability. Pay specific attention to punctuations and vague sentences. Lines 469-470 “To corroborate this data, a treatment with a blend of different natural compounds including CoQ, astaxanthin and others…” Data is a plural term. It should be “these data” and not “this data”. What do astaxanthin and “others” have in common with CoQ that leads to their potentiating or synergistic effect on triglycerides? Please elaborate on this and provide references.
4.4. CoQ deficiency syndrome
The entire section must undergo significant proofreading to improve the quality of presentation and the readability of the manuscript. Lines 498-500 “In general, patients show symptoms related to a specific organ dysfunction or the pathology could affect more than one tissue resulting in a multiorgan disorder…” The meaning of this sentence is unclear. Revise and simplify. Line 501, “On the other hand” is not the right transition here. Line 504, was “oxphos” defined in the text? It has been used more than once.
5. Challenges and conclusions
Lines 525-526 “Regarding CoQ, it exerts a fundamental role in the maintenance of cellular homeostasis being involved both in the control of ETC and in the cellular antioxidant system …” This sentence reads awkward and must be revised. Line 537, what does “Until know we can…” mean?
Author Response
Reviewer 3
General comments:
The manuscript titled “CoQ10 and resveratrol effects to ameliorate aged-related mitochondrial dysfunctions” by Gherardi et al. provides a compendium of the latest advances on the potential role of the mitochondrial electron carrier CoQ10 and the reactive oxygen and nitrogen species scavenger resveratrol in mitigating age-related mitochondrial dysfunction. The premise of the work is that “any treatment…to improve and sustain mitochondrial functionality is a good candidate to counteract mitochondrial-associated dysfunction” during aging. In this review, the authors provided insights into the use of these compounds in recent clinical trials and metanalysis as it relates to mitochondrial changes during aging. The authors endeavored to provide a balance coverage of the matter, highlighting controversies where possible. In addition, the authors presented potential cautionary factors to be considered when considering experimental and clinical cases, and dose and duration in administering the supplements. They are commended for this diligent and masterful effort. However, my enthusiasm for the work was significantly diminished by the low quality of the presentation. It was disappointing to see the quality of writing in this manuscript to be below what one would expect for a reputable journal. The entire manuscript was challenging and laborious to read. Unfortunately, these negative attributes overshadowed what I initially thought would be an interesting topic to read. There are sentences that lack punctuations in the right place or contain misplaced punctuations, and spelling errors or use of wrong words. Some sentences are vague and awkward. This makes for a tough reading. The authors are highly recommended to seek professional help before resubmitting a revision, i.e., if they are accorded the opportunity. Any future resubmission should be accompanied by a certificate of authentication, showing proof of professional editing assistance. There is also lack of consistency in the use of tenses, past and present, and this also contributes towards the complicated nature of the manuscript. It will take an inordinate amount of time to pinpoint all the flaws and grammatically challenging phrases in this manuscript. In this critique, only highlights of some glaring ones will be noted. The onus for good presentation is on the authors. They should do a thorough job polishing and fine-tuning their paper before submission. Lastly, the authors must avoid using this phrase “thanks to the…” repeatedly.
We agree with the reviewer’s comment. Thus, we have extensively reviewed the text regarding contents, grammar, and syntax. Regarding the first two sections, the main differences are highlighted in red. In addition, the CoQ section has been changed.
Specific comments
Abstract:
Line 19 “…function during the elderly…” Revise to read ““…function during the aging process…” There several areas in the manuscript where similar phrases are used. These should all be corrected. Lines 19-20 “In the last years interest is growth on natural compounds able to modulate mitochondrial function…” is an awkward sentence and must be revised. Lines 21-23 “…we will discuss about recent clinical trials and meta-analyses based on resveratrol and CoQ supplementation during elderly…” is another sentence that must be revised for clarity of meaning. There are other minor syntax errors that the authors must address.
We corrected the sentences.
- Introduction
Lines 28-30 “…in that case, glycolytic products could be oxidized in a mitochondrial-independent way in the cytosol whit a less efficient anaerobic respiration” This sentence could be simplified for better understanding. Lines 42-44 “The ETC is the major reactive oxygen species (ROS) producer, indeed, although oxidative phosphorylation is an efficient process, a small amount of electrons leaks from the ETC and interacts with oxygen forming ROS…” Revise to read, “The ETC is the major source for reactive oxygen species (ROS) production; indeed, although oxidative phosphorylation is an efficient process, a small amount of electrons leaks from the ETC and interacts with molecular oxygen to produce ROS…” Line 45, “aging” should be added at the end of the phrase “…”free radical theory…” Lines 51-54 “Although mitochondria play a central role in the control of energy metabolism, today we can no longer limiting our view only on their energetics behaviors, but rather mitochondria are platforms for many intracellular signals interfering in many different physiological and pathological settings.” This entire sentence is complex and reads awkward. It must be revised. In this section, other issues, such as poor punctuations, abound. The entire section must undergo thorough review.
We corrected the sentences, and we have extensively reviewed the introduction.
- Mitochondrial alterations during aging
Line 69 “Several studies show that aging is accompanied by a drop of mitochondrial function…” Revise to read, “Several studies show that aging is associated with a decrease in mitochondrial function…” This sentence, lines 91-92, “In conclusion, these discrepancies could be combine together considering ROS a necessary survival signal induced by different stress conditions with the scope of ameliorate the progressive…” is grammatically flawed and must be revised. This sentence, lines 97-99, “On the other hand, in elderly ROS production increases overcoming the levels required for the proper signaling function, eventually magnifying, instead of being beneficial, the aging condition…” is tough to decipher because of the poor structure of the sentence. Line 111 “…progressively acquires different features of premature aging such as alopecia…” reads awkward. Consider revising line 128 “Mitochondrial alterations during aging are also linked to mitohormesis, meant as the…” is a grammatically poor sentence and must be revised. In the sentence, lines 131-133, what does “…different compounds such as metformin and resveratrol are mild mitochondrial poisons…mean, even though they can activate AMPK? Lines 139-141 “Until now, the best intervention…stimulating mitochondrial biogenesis…” should be revised. In addition to these comments, there are other syntax errors and bad punctuations that must be corrected. This will improve the readability and understanding of the sentence.
We corrected the sentences, and we extensively reviewed this section
- Resveratrol
3.1. Chemistry, safety and bioavailability of resveratrol
Lines 146-147 “In detail, resveratrol is characterized by two phenolic rings with a double bound, thus resveratrol…” Revise to read, “The structure of resveratrol is characterized by two phenolic rings with double bonds, thus resveratrol…” Line 151, what do the authors mean by “…the oxidative stress should be strictly tuned in the cell…”? Line 185 “In this contest…” It is unclear what the authors are trying to say here. “Contest” does not seem to be the right word here. Please explain. In the same sentence, what does the phrase “…mitochondria are at the center of its beneficial effect…” mean? Line 200 “…it should be taken into account that resveratrol is able to inhibit the F0F1-ATPase activity in a concentration-dependent manner…” Simplify to read, “…it should be noted that resveratrol inhibits the F0F1-ATPase activity in a concentration-dependent manner…” The paragraph, lines 203-213, needs a significant revision, including use of proper punctuations and the proper use of the article “the”. Similarly, the entire paragraph, lines 219-236, must be revised. It has awkward phrases, poor syntax and improper punctuations. These issues make it challenging for one to read and understand the text. For example, what does the phrase “…resveratrol could have an indirect antioxidant effect based on its capacity to modulate the expression of proteins induced in during scavenger activity…” mean?
We corrected the sentences, and we extensively reviewed this section.
3.3. Beneficial effects of resveratrol during aging and aged-associated diseases
Just like the previous sections, this entire section also requires careful scrutiny of the language structure. Too many tough sentences/phrases and unclear meanings. Line 243 “Since the first evidence that resveratrol could act mimicking some anti-aged effects of dietary restriction, multiple studies have been accumulated demonstrating the positive…” What does this sentence mean? The sentence, “Different studies demonstrated that resveratrol could counteracts this situation impinging on mitochondrial homeostasis…” must be revised and simplified. Line 260, what does this “…it also modulates the sarcomere structure impinging on AMPK pathway…” mean? Line 264 “Conflicting results have emerged in human studies…” What are the “conflicting results”? Line 271 “Besides skeletal muscle also cardiovascular functionality, in terms of endothelial dysfunction and alteration in the redox balance, is involved in age-related…” Simplify. The paragraph, lines 292-308 contain unusual sentences, poor syntax and improper use of punctuations. Difficult to understand the message being conveyed. Lines 317-318 “These assumption indicate resveratrol as a promising agent against obesity and diabetes…” is a flawed sentence. Lines 321-322 “Concerning human studies, even if controversial data are present, it is clear that the efficacy of resveratrol treatment depends on the metabolic status…” What does this mean? Revise. Lines 324-326 “…as an adjunct to the canonical pharmacological therapy show that this molecule is able to improve some cardiometabolic markers but not glycemia, insulin levels and lipid metabolism…” This is a complicated sentence to decipher, and meaning is unclear. In the phrase, “Finally, gestational diabetes, not less important than T2DM…”, why is this information comparing the importance of the two types of diabetes necessary here to mention here?
We corrected the sentences, and we extensively reviewed this section.
- CoenzymeQ10
4.1. Chemistry of CoQ10
The entire paragraph, lines 339-356, must undergo extensive review for proper grammar and syntax use. What is meant by “by a polar head bound”? “It is worth to mention that the reduced form of CoQ can be regenerated at different cell location, not only into the mitochondria.” A reference/s must be cited here. Lines 357-363 should include some citations, to verify some of the claims made.
See comment below
4.2. CoQ10 as a key factor to control cellular homeostasis
This entire section must undergo stringent reevaluation for syntax, grammar and presentation.
Line 365 “In this section we will focused…” This is an astounding and glaring grammatical error.
See comment below
4.3. Beneficial effects of CoQ10 during aging and age-associated diseases
This section should be revised to improve its readability. Pay specific attention to punctuations and vague sentences. Lines 469-470 “To corroborate this data, a treatment with a blend of different natural compounds including CoQ, astaxanthin and others…” Data is a plural term. It should be “these data” and not “this data”. What do astaxanthin and “others” have in common with CoQ that leads to their potentiating or synergistic effect on triglycerides? Please elaborate on this and provide references.
See comment below
4.4. CoQ deficiency syndrome
The entire section must undergo significant proofreading to improve the quality of presentation and the readability of the manuscript. Lines 498-500 “In general, patients show symptoms related to a specific organ dysfunction or the pathology could affect more than one tissue resulting in a multiorgan disorder…” The meaning of this sentence is unclear. Revise and simplify. Line 501, “On the other hand” is not the right transition here. Line 504, was “oxphos” defined in the text? It has been used more than once.
Regarding section 4, it has been wholly rewritten, considering all the reviewer’s comments.
- Challenges and conclusions
Lines 525-526 “Regarding CoQ, it exerts a fundamental role in the maintenance of cellular homeostasis being involved both in the control of ETC and in the cellular antioxidant system …” This sentence reads awkward and must be revised. Line 537, what does “Until know we can…” mean?
We revised the awkward sentence, and we corrected the syntax.
Reviewer 3 Report
Dear Authors
This manuscript is well written and can be accepted after minor revision.
This is a review article. Thus, a few tables will be helpful.
Thus, I recommend the table to read and figure it out easily.
1) Please, make a Table regarding "Beneficial effects of resveratrol during aging and aged-associated diseases"
2) Please, make a Table regarding "Beneficial effects of CoQ10 during aging and age-associated diseases"
3) The image in Figure 1 is not good, and authors need to improve this image.
Author Response
Dear Authors
This manuscript is well written and can be accepted after minor revision.
This is a review article. Thus, a few tables will be helpful.
Thus, I recommend the table to read and figure it out easily.
1) Please, make a Table regarding "Beneficial effects of resveratrol during aging and aged-associated diseases"
2) Please, make a Table regarding "Beneficial effects of CoQ10 during aging and age-associated diseases"
3) The image in Figure 1 is not good, and authors need to improve this image.
We thank the reviewer for these valuable indications. We added the two tables and we improve figure 1 (Figure 2 in the new version).
Round 2
Reviewer 1 Report
Authors have revised this version addressing the reviewers comments.
Author Response
We thank the reviewer.
Reviewer 2 Report
The manuscript presentation has significantly improved. The authors have taken some of my suggestions into consideration. Consequently, the readability seems smoother than the original submission. However, some sentences are still vague or unclear. These are located in the areas where the authors do not show their corrections highlighted in red font.
Section 4.1:
Line 354 "Although CoQ10 can be introduced with the diet..." It is not clear here whether the authors meant CoQ10 can be found in the diet or can be added as a supplement to the diet.
Section 4.3:
Line 408, what does "CoQ levels and exercise correlation during aging is another crucial aspect..." mean? Lines 428-429, this sentence "In addition, CoQ treatment prevents some complications, such as arrhythmias, in patients undergoing to cardiac surgery..." should be revised. It reads awkward. Lines 431-433 "Interestingly, patients suffering from chronic heart failure have been subjected to 2-years CoQ treatment, demonstrating a reduction in major cardiovascular events..." The tense of this sentence is not proper. Revise. Line 439 "uncoherent" is not a word. It is "incoherent". Line 440 "On the one hand..." is not the appropriate transitional phrase here. The authors have used this phrase, "On the other hand" excessively in this manuscript.
Section 4.4:
Lines 467-468, this sentence "Since the key role exerted by CoQ in the mitochondria, its reduction leads to the impairment of oxidative phosphorylation with the final decrease in ATP production..." is incomplete. Line 479 "oxphos". The full spelling "oxidative phosphorylation" should be used instead of this unorthodox abbreviation.
Section 5
This section should be read carefully to wilt out awkward sentences. Line 485 "whit". Although "whit" is a word, which means a small amount, it is not the right word here. Revise.
Author Response
The manuscript presentation has significantly improved. The authors have taken some of my suggestions into consideration. Consequently, the readability seems smoother than the original submission. However, some sentences are still vague or unclear. These are located in the areas where the authors do not show their corrections highlighted in red font.
We thank the reviewer for the comments. We revised this version of the manuscript accordingly to his suggestions.
Section 4.1:
Line 354 "Although CoQ10 can be introduced with the diet..." It is not clear here whether the authors meant CoQ10 can be found in the diet or can be added as a supplement to the diet.
We rephrased the sentence to be more precise.
Section 4.3:
Line 408, what does "CoQ levels and exercise correlation during aging is another crucial aspect..." mean? Lines 428-429, this sentence "In addition, CoQ treatment prevents some complications, such as arrhythmias, in patients undergoing to cardiac surgery..." should be revised. It reads awkward. Lines 431-433 "Interestingly, patients suffering from chronic heart failure have been subjected to 2-years CoQ treatment, demonstrating a reduction in major cardiovascular events..." The tense of this sentence is not proper. Revise. Line 439 "uncoherent" is not a word. It is "incoherent". Line 440 "On the one hand..." is not the appropriate transitional phrase here. The authors have used this phrase, "On the other hand" excessively in this manuscript.
We revised the awkward sentences and substituted "on the other hand" with a more appropriate adverb.
Section 4.4:
Lines 467-468, this sentence "Since the key role exerted by CoQ in the mitochondria, its reduction leads to the impairment of oxidative phosphorylation with the final decrease in ATP production..." is incomplete. Line 479 "oxphos". The full spelling "oxidative phosphorylation" should be used instead of this unorthodox abbreviation.
Oxphos is a well-established abbreviation for oxidative phosphorylation, which is full-spelled at line 42. However, as suggested by the reviewer, we used the full name where appropriate.
Section 5
This section should be read carefully to wilt out awkward sentences. Line 485 "whit". Although "whit" is a word, which means a small amount, it is not the right word here. Revise.
We revised the entire section.